# Multi-Agent Decision-Making Modes in Uncertain Interactive Traffic Scenarios via Graph Convolution-Based Deep Reinforcement Learning

**DOI:** 10.3390/s22124586

**Published:** 2022-06-17

**Authors:** Xin Gao, Xueyuan Li, Qi Liu, Zirui Li, Fan Yang, Tian Luan

**Affiliations:** 1School of Mechanical Engineering, Beijing Institute of Technology, Beijing 100080, China; 3120210298@bit.edu.cn (X.G.); 3120195255@bit.edu.cn (Z.L.); yangfanbitdb@163.com (F.Y.); 3220200285@bit.edu.cn (T.L.); 2Department of Transport and Planning, Faculty of Civil Engineering and Geosciences, Delft University of Technology, Stevinweg 1, 2628 CN Delft, The Netherlands

**Keywords:** multi-mode decision-making, connected autonomous vehicles, reward function matrix, uncertain highway exit scene, GQN, MDGQN

## Abstract

As one of the main elements of reinforcement learning, the design of the reward function is often not given enough attention when reinforcement learning is used in concrete applications, which leads to unsatisfactory performances. In this study, a reward function matrix is proposed for training various decision-making modes with emphasis on decision-making styles and further emphasis on incentives and punishments. Additionally, we model a traffic scene via graph model to better represent the interaction between vehicles, and adopt the graph convolutional network (GCN) to extract the features of the graph structure to help the connected autonomous vehicles perform decision-making directly. Furthermore, we combine GCN with deep Q-learning and multi-step double deep Q-learning to train four decision-making modes, which are named the graph convolutional deep Q-network (GQN) and the multi-step double graph convolutional deep Q-network (MDGQN). In the simulation, the superiority of the reward function matrix is proved by comparing it with the baseline, and evaluation metrics are proposed to verify the performance differences among decision-making modes. Results show that the trained decision-making modes can satisfy various driving requirements, including task completion rate, safety requirements, comfort level, and completion efficiency, by adjusting the weight values in the reward function matrix. Finally, the decision-making modes trained by MDGQN had better performance in an uncertain highway exit scene than those trained by GQN.

## 1. Introduction

Artificial intelligence is in its golden development age due to the exponential increase in data production and the continuous improvements in computing power [1]. Autonomous driving is one of the main uses. A comprehensive autonomous driving system integrates sensing, decision-making, and motion-controlling modules [2,3,4]. As the “brains” of connected autonomous vehicles (CAVs) [5], the decision-making module formulates the most reasonable control strategy according to the state feature matrix transmitted by the sensing module, the vehicle state, and the cloud transmission information [6]. Moreover, it sends the determined control strategy to the motion-controlling module, including high-level behavior and low-level control requirements [7,8]. It is crucial to complete autonomous driving tasks safely and efficiently by making reasonable decisions based on other modules [9].

In an uncertain interactive traffic scenario, the driving environment has rigorous dynamic characteristics and high uncertainty, and the influences of driving behaviors of different traffic participants will be transmitted continuously [10]. On the level of transportation overall, all traffic participants need to cooperate efficiently [11]. At the traffic participant level, individuals need to judge the risk factors and make appropriate decisions sensitively based on dynamic scene changes [12]. In [13], a Gaussian mixture model and important weighted least squares probability classifier were combined and used for scene modeling. That model can identify the braking strength levels of new drivers under the condition of insufficient driving data. The key to participating in traffic scenarios for CAVs is that each CAV needs to generate appropriate and cooperative behavior to match human vehicles and other CAVs. Therefore, CAVs urgently demand efficient and accurate multi-agent decision-making technology to effectively handle the interactions between different traffic participants.

The current multi-agent decision-making technologies mainly focus on deep reinforcement learning (DRL) due to the excellent performance of DRL in high-dimensional dynamic state space [14]. The keys of DRL in uncertain interactive traffic scenarios can be summarized as follows: (1) Efficient modeling of interactive traffic scenes and accurate representation of state features. (2) Generating reasonable and cooperative decision-making behaviors based on uncertain scene changes and individual task requirements. The design of the reward function is an essential part of the DRL application. Concretizing and numericizing task objectives realizes the communication between objectives and algorithms. Therefore, the design of the reward function determines whether the agent can generate reasonable and cooperative decision-making behaviors [15]. In addition, the accurate modeling of the interactive traffic environment and representation of state characteristics are the requirements for agents to generate expected behaviors.

In studies of traffic scenarios using the DRL method, researchers found that in uncertain interactive traffic scenarios, sparse reward problems lead to agents having a lack of effective information guidance [16], making the algorithm difficult to converge. In order to solve the sparse reward problem, researchers divide the original goal into different sub-goals and give reasonable rewards or punishments. In [17], the DDPG was adopted to settle the autonomous braking problem. The reward function was split into three parts to solve the problems: braking too early, braking too late, and braking too quickly. In [18], the reward function was divided into efficiency and safety rewards to train the comprehensive safety and efficiency decision model. In addition, considering the changes in driving task requirements and scene complexity, some studies have given different weights to the sub-reward functions based on the decomposition of the reward function, so as to train different decision-making modes. In [19], the reward function was divided into sub-reward functions based on security, efficiency, and comfort. The system can realize different control targets by adjusting the weight values in the reward function.

In the decision-making research involving uncertain interactive traffic scenarios, the DRL method only takes the individual characteristics of each vehicle as the input. It ignores the interactive influence of transitivity between vehicles. This will result in CAVs not generating reasonable and cooperative behavior, which may reduce total traffic efficiency and the occurrence of traffic accidents. Graph representation can accurately describe the interactions between agents, representing the relationship between vehicles in uncertain interactive traffic scenarios. Therefore, some researchers focused on the graph reinforcement learning (GRL) method and modeled the interaction with graph representation [20]. In [21], a hierarchical GNN framework was proposed and combined with LSTM to model the interactions of heterogeneous traffic participants (vehicles, pedestrians, and riders) to predict their trajectories. The GRL method combines GNN and DRL: the features of interactive scenes are processed by GNN, and cooperative behaviors are generated by the DRL framework [22]. In [23], the traffic network was modeled by dynamic traffic flow probability graphs, and a graph convolutional policy network was used in reinforcement learning.

This paper proposes an innovative, dynamic reward function matrix, and various decision-making modes can be trained by adjusting the weight coefficient of the reward function matrix. Additionally, the weight coefficient is further set as a function of reward, forming an internal dynamic reward function. In the traffic environment adopted in this paper, the randomness and interactions between HVs and CAVs are strengthened by using some human vehicles making uncertain lane changes. Two GRL algorithms are used in this paper, GQN and MDGQN. Finally, we report a simulation based on the SUMO platform and a comparative analysis from various perspectives, such as reward functions, algorithms, and decision-making modes. The schematic diagram of the designed framework is shown in Figure 1.

To summarize, the contributions of this paper are as follows:1.Innovative dynamic reward function matrix: we propose a reward function matrix including a decision-weighted coefficient matrix, an incentive-punished-weighted coefficient matrix, and a reward–penalty function matrix. By adjusting the decision-weighted coefficient matrix, the decision-making modes of different emphases among driving task, traffic efficiency, ride comfort, and safety can be realized. Based on the premise that the incentive-punished function matrix separates the reward and the penalty, the optimization of individual performance can be achieved by adjusting the incentive and punishment ratio of each sub-reward function. In addition, the weight coefficient of the incentive-punished-weighted coefficient matrix is further set as a function of reward functions, which can reduce the impact of proportional adjustment on important operations, such as collision rate.2.Adjust the parameters to train multiple decision-making modes: We compare the proposed reward function matrix with the traditional reward function under the same conditions. By adjusting the parameters of the decision-weighted coefficient matrix and the incentive-punished-weighted coefficient matrix, we can achieve aggressive or conservative incentive and punitive decision-making modes, respectively. Specifically, the four decision-making modes trained in this paper are the aggressive incentive (AGGI), aggressive punishment (AGGP), conservative incentive (CONI), and conservative punishment (CONP).3.Modeling of interactive traffic scene and evaluation of decision-making modes: We designed a highway exit scene with solid interactions between CAVs and human vehicles (HVs) and adopted two algorithms to verify their differences. Additionally, we also propose a set of indicators to evaluate the performance of driverless decision-making and used them to verify the performance differences among various algorithms and decision-making modes.

This article is organized as follows. Section 2 introduces the problem formulation. Section 3 introduces the methods used. Section 4 proposes the reward function matrix. Section 5 describes and analyzes the simulation results. Section 6 summarizes this paper and gives the future development directions.

## 2. Problem Formulation

### 2.1. Scenario Construction

As is shown in Figure 2, a 3-lane highway containing two exits was designed, and the three lanes were sorted from bottom to top as first, second, and third lanes. All the vehicles (HVs and CAVs) enter the road segment from the left of “Highway 0”. The color of vehicle indicates its type and intention. In this paper, the vehicles were set as follows for safety principles and actual human driving rules:

A white body with a colored roof represents an HV, and an all-blue vehicle represents a CAV;The HV with the red roof (HV1) is set to appear in three lanes randomly and can only go out from “Highway 2”;The HV with the orange roof (HV2) is set to emerge from the second lane and can go out from “Exit 1“ when it is in the first lane and go out from “Highway 2” when it is in the other lanes;The HV with the green roof (HV3) is set to only appear from the first lane and can only go out from “Exit 0”;When there is no car at the longitudinal distance of 10 m in the lane to be changed, the HV2 will be switched to the right;The CAV is set to appear in three lanes randomly and can go out from “Highway 2”, “Exit 0”, and “Exit 1”.

The interaction between CAVs and HVs is enhanced through the above setting. The setting of HV1 enhances the uncertainty of the scene considerably, increasing the requirements for CAVs’ decision-making. Straightening away from “Highway 2” is the most straightforward choice with the lowest collision risk, owing to CAVs’ least lane change decisions. In addition, due to the uncertain lane change behavior of HV2, it is difficult for CAVs to explore “Exit 1” safely, and the collision risk of leaving the highway from “Exit 1” is higher than it is for other exits. Specific scenario setting parameters are shown in Table 1:

### 2.2. State Representation

The scenario is modeled as an undirected graph. Each vehicle in this scenario is regarded as the node of the graph, and the interaction between vehicles is considered the edge of the graph. Three matrices can represent the state space: a node features matrix $X_{t}$, an adjacency matrix $A_{t}$, and a CAV mask matrix $M_{t}$; each of them are described in the following.

Node Features Matrix: The vehicle’s features are speed, position, location, and intention, denoted as $\left[V_{i},Y_{i},L_{i},I_{i}\right]$. The node features matrix represents the features of each vehicle in the constructed scenario, which can be described as follows:(1)$$N_{t}=\left[\begin{array}{c} \left[V_{1},Y_{1},L_{1},I_{1}\right]\\ \left[V_{2},Y_{2},L_{2},I_{2}\right]\\ \cdot \cdot \cdot \\ \left[V_{i},Y_{i},L_{i},I_{i}\right]\\ \cdot \cdot \cdot \\ \left[V_{n},Y_{n},L_{n},I_{n}\right] \end{array}\right]$$
where Vi=vi−actualvi−actualvmaxvmax denotes the ratio of actual longitudinal velocity to maximum longitudinal velocity. Yi=yi−actualyi−actualLhighwayLhighway denotes the percentage of actual longitudinal position of the total length of the highway. $L_{i}$ denotes the one-hot encoding matrix of the current lane of vehicles. $I_{i}$ denotes the vehicle’s one-hot encoding matrix of current intention (change to left lane, change to right lane, and go straight).

Adjacency matrix: In this paper, the interactions between vehicles are embodied in the information sharing between vehicles, expressed by the adjacency matrix. The calculation of the adjacency matrix is based on three assumptions:All CAVs can share information in the constructed scenario;Information cannot be shared between HVs;Vehicles can share information with themselves $a_{ii}=1$.All CAVs can share information with HVs in their sensing range.

The derivation of the adjacency matrix is as follows:(2)$$A_{t}=\left[\begin{array}{cccccc} a_{11} & a_{12} & \cdots & & \cdots & a_{1n}\\ a_{21} & a_{22} & \cdots & & \cdots & a_{2n}\\ \vdots & \vdots & \ddots & & & \vdots \\ & & & a_{ij} & & \\ \vdots & \vdots & & & \ddots & \vdots \\ a_{n1} & a_{n2} & \cdots & & \cdots & a_{nn} \end{array}\right]$$
where $a_{ij}$ denotes the edge value of the ith vehicle and the jth vehicle. $a_{ij}=1$ denotes that the ith vehicle and the jth vehicle share information; $a_{ij}=0$ denotes that the ith vehicle and the jth vehicle share no information.

CAV mask matrix: CAV mask is used to filter out the embeddings of HVs after the GCN fusion block. The specific mathematical expression is as follows:(3)$$M_{t}=\left[m_{1},m_{2},\cdots ,m_{i},\cdots m_{n}\right]$$
where $m_{i}$ = 0 or 1. If the ith vehicle is controlled by the GRL algorithm, $m_{i}=1$; otherwise, $m_{i}=0$.

### 2.3. Action Space

At each time step, each CAV has a discrete action space representing potential actions to be executed at the next time step, as follows: (4)$$a_{i}=\left\{a_{lane-change},a_{acceleration}\right\}$$
where $a_{lane-change}$ indicates that the lane change action can be taken, including a left lane change, right lane change, or straight line; $a_{acceleration}$ denotes the discrete acceleration that CAV can take, and its value is equalized by interval $[-8\;m\cdot s^{-2},5\;m\cdot s^{-2}]$ at 0.5 $m\cdot s^{-2}$.

## 3. Methods

This section describes the principles of the methods used, including graph convolutional neural networks, Q-learning, GQN, and MDGQN.

### 3.1. Graph Convolutional Neural Network

GCN is a neural network model that directly encodes graph structure. The goal is to learn a function of features on a graph. A graph can be represented by G=(V,E) in theory, where *V* denotes the set of nodes in the graph, the number of nodes in the graph is denoted by *N*, and *E* denotes the set of edges in the graph. The state of *G* is considered a tuple of 3 matrices of information: feature matrix *X*, adjacency matrix *A*, and degree matrix *D*.

The adjacency matrix *A* is used to represent connections between nodes;The degree matrix *D* is a diagonal matrix, and $D_{ii}=\sum_{j}A_{ij}$;The feature matrix *X* is applied to represent node features, $X\in R^{N\times F}$, where *F* represents the dimensions of the feature.

GCN is a multi-layer graph convolution neural network, a first-order local approximation of spectral graph convolution. Each convolution layer only deals with first-order neighborhood information, and multi-order neighborhood information transmission can be realized by adding several convolution layers.

The propagation rules for each convolution layer are as follows [24]:(5)$$Z=g\left(H,A\right)=\sigma \left({\hat{D}}^{-1/2}\hat{A}{\hat{D}}^{-1/2}H^{\left(l\right)}W^{\left(l\right)}+b\right)$$
where $\hat{A}=A+I_{N}$ is the adjacency matrix of the undirected graph *G* with added self-connections. $I_{N}$ is the identity matrix; ${\hat{D}}_{ii}=\sum_{j}{\hat{A}}_{ij}$ and $W^{\left(l\right)}$ are layer-specific trainable weight matrices. σ(·) denotes an activation function, such as the ReLU(·)=max(0,·). $H^{\left(l\right)}\in R^{N\times D}$ is the matrix of activations in the lth layer, H(0)=X.

### 3.2. Deep Q-Learning

Q-learning [25] is a value-based reinforcement learning algorithm. $Q_{t}$ is the expectation that $Q(s_{t},a_{t})$ can obtain benefits by taking action $a_{t}\left(a_{t}\in A_{t}\right)$ under the state of $s_{t}\left(s_{t}\in S_{t}\right)$ at time *t*. The environment will feed back the corresponding reward $R_{t}$ according to the agent’s action. Each time step produces a quadruplet ($s_{t},a_{t},r_{t},s_{t+1}$), and it is stored in the experience replay. Deep Q-learning [26] replaces the optimal action–value function with the deep neural network Q(s,a,ω). The following is a description of the principle of the algorithm.

The predicted values of DQN can be calculated according to the given four-tuple ($s_{t},a_{t},r_{t},s_{t+1}$): (6)$${\hat{q}}_{t}=Q\left(s_{t},a_{t},\omega \right)$$

The TD target can be calculated based on the actual observation reward $r_{t}$: (7)$${\hat{y}}_{t}=r_{t}+\gamma \cdot \max_{a\in A}Q\left(s_{t+1},a,\omega \right)$$

DQN updates network parameters according to the following formula:(8)$$\omega \leftarrow \omega -\alpha \cdot \left({\hat{q}}_{t}-{\hat{y}}_{t}\right)\cdot \nabla_{\omega }Q\left(s_{t},a_{t},\omega \right)$$
where α represents the learning rate.

### 3.3. Graph Convolutional Q-Network (GQN)

As described in Section 3.2, Q-learning uses a Q-Table to store the Q value of each state–action pair. However, if the state and action space are high-dimensionally continuous, there will be the curse of dimensionality; that is, with a linear increase in dimensions, the calculation load increases exponentially. GQN [27] replaces the optimal action–value function $Q_{⋆}\left(s,a\right)$ with the graph convolutional neural network Q(s,a,θ):(9)Q(s,a,θ)=ρ(Z,a)
where *Z* is the node embeddings output from graph convolution layer. ρ represents the neural network block, including the fully connected layer. θ is the aggregation of all the weights.

The specific training process of GQN is the same as DQN [26]. Firstly, the predictive value of GQN can be calculated according to the four-tuple ($s_{t},a_{t},r_{t},s_{t+1}$) sampled from the experience replay:(10)$${\tilde{q}}_{t}=Q\left(s_{t},a_{t},\theta_{now}\right)$$

However, since the Q-network uses the same estimate to update itself, it will cause bootstrapping and lead to deviation propagation. Therefore, another neural network can be used to calculate the TD target, called the target network $Q(s,a,\theta_{now}^{-})$. Its network structure is precisely the same as that of the Q-network, but the parameter $\theta_{now}^{-}$ is different from $\theta_{now}$. Selecting the optimal action and forward propagation of the target network: (11)$$a^{⋆}=\underset{a\in A}{\arg\max}Q\left(s_{t+1},a_{t},\theta_{\mathrm{now}}^{-}\right)$$
(12)$${\tilde{q}}_{t+1}=Q\left(s_{t+1},a^{⋆},\theta_{now}^{-}\right)$$

Calculation of TD target ${\hat{y}}_{t}$ and TD error $\delta_{t}$: (13)$${\hat{y}}_{t}=r_{t}+\gamma \cdot {\tilde{q}}_{t+1}$$
(14)$$\delta_{t}={\tilde{q}}_{t}-{\hat{y}}_{t}$$

The gradient $\nabla_{\theta }Q\left(s_{t},a_{t},\theta_{now}\right)$ is calculated by the backpropagation of a Q-Network, and the parameter of the Q-Network is updated by gradient descent: (15)$$\theta_{new}\leftarrow \theta_{now}-\alpha \cdot \delta_{t}\cdot \nabla_{\theta }Q\left(s_{t},a_{t},\theta_{now}\right)$$
where $\theta_{new}$ is the parameter of the updated Q-Network. α represents the learning rate.

Finally, GQN adopts the weighted average of the two networks to update the target network parameters: (16)$$\theta_{new}^{-}\leftarrow \tau \cdot \theta_{new}+\left(1-\tau \right)\cdot \theta_{now}^{-}$$
where τ represents soft update rate.

### 3.4. Multi-Step Double Graph Convolutional Q-Network (MDGQN)

MDGQN further adopts double Q-learning and a multi-step TD target algorithm based on GQN. As shown in Section 3.3, the target network cannot completely avoid bootstrapping, since the parameters of the target network are still related to the Q-Network. The double Q-learning algorithm is improved based on the target network. The Q-learning with the target network uses the target network to select the optimal action and calculate the Q value of the optimal action. However, the double Q-learning selects the optimal action according to the Q-Network and uses the target network to calculate the Q value of the optimal action. Equation (Equation 11) is modified as follows: (17)$$a^{⋆}=\underset{a\in A}{\arg\max}Q\left(s_{t+1},a_{t},\theta_{\mathrm{now}}\right)$$

The multi-step TD target algorithm can balance the significant variance caused by Monte Carlo and the significant deviation caused by bootstrapping. Equation (Equation 13) is modified as follows: (18)$${\hat{y}}_{t}=\sum_{i=0}^{m-1}{\gamma^{i}r}_{t+i}+\gamma^{m}\cdot {\tilde{q}}_{t+m}$$
where ${\hat{y}}_{t}$ is called the m-step TD target.

## 4. Reward Functions

The design of the reward function is an important criterion and goal of the DRL training process. Based on four aspects—the results the of the driving task, traffic efficiency, ride comfort, and safety—the reward function is divided into four blocks. Further, we propose a reward function matrix including a decision-weighted coefficient matrix, an incentive-punished-weighted coefficient matrix, and a reward–penalty function matrix.
(19)$$\begin{array}{c} r=tr(A_{k1}A_{k2}A_{r})\\ =tr\left[\left[\begin{array}{cccc} k_{r} & & & \\ & k_{e} & & \\ & & k_{c} & \\ & & & k_{s} \end{array}\right]{\left[\begin{array}{cccc} k_{rI} & & & \\ k_{rP} & & & \\ & k_{eI} & & \\ & k_{eP} & & \\ & & k_{cI} & \\ & & k_{cP} & \\ & & & k_{sI}\\ & & & k_{sP} \end{array}\right]}^{T}\left[\begin{array}{cccc} r_{result-I} & & & \\ r_{result-P} & & & \\ & r_{efficiency-I} & & \\ & r_{efficiency-P} & & \\ & & r_{comfort-I} & \\ & & r_{comfort-P} & \\ & & & r_{safe-I}\\ & & & r_{safe-P} \end{array}\right]\right] \end{array}$$

Specific parameters are described below:1.$A_{k1}$ is the decision-weighted coefficient matrix. $k_{r}$, $k_{e}$, $k_{c}$, and $k_{s}$ denote the weights of reward and penalty functions based on the results of the driving task, traffic efficiency, ride comfort, and safety.2.$A_{k2}$ is the incentive-punished-weighted coefficient matrix. $k_{rI}$, $k_{eI}$, $k_{cI}$, and $k_{sI}$ denote the weights of reward functions based on the results of the driving task, traffic efficiency, ride comfort, and safety, respectively. $k_{rP}$, $k_{eP}$, $k_{cP}$, and $k_{sP}$ denote the weights of penalty functions based on the results of the driving task, traffic efficiency, ride comfort, and safety, respectively.3.$A_{r}$ is a reward–penalty function matrix. $r_{result-I}$, $r_{efficiency-I}$, $r_{comfort-I}$, and $r_{safe-I}$ are reward functions based on the results of the driving task, traffic efficiency, ride comfort, and safety, respectively. $r_{result-P}$, $r_{efficiency-P}$, $r_{comfort-P}$, and $r_{safe-P}$ are penalty functions based on the results of the driving task, traffic efficiency, ride comfort, and safety, respectively.

By adjusting the weight coefficient of the decision-weighted coefficient matrix and incentive-punished-weighted coefficient matrix, DRL can train different goal-oriented decision-making modes. In the decision-making process of autonomous vehicles, the upper control module can choose different decision-making modes according to different needs. In order to select a decision-making model with excellent comprehensive performance and strong contrast, we conducted multiple sets of experiments, and some experimental data were put in Appendix A. This paper determined four decision modes: AGGI, AGGP, CONI, and CONP, by adjusting the parameters in Table 2 and Table 3.

Since the change of weight coefficient will dilute some essential rewards or punishments, this paper improves the reward function against this defect. The weight coefficient of the incentive-punished-weighted coefficient matrix is further set as a functional of reward functions, which forms an internal dynamic reward function. The specific formula is as follows: (20)$$\left\{\begin{array}{c} k_{rI}=k_{rI0}\cdot \exp\left((r_{result-P}+r_{comfort-P})/100,000\right)\\ k_{eI}=k_{eI0}\cdot \exp\left((r_{result-P}+r_{comfort-P})/80,000\right)\\ k_{sI}=k_{sI0}\cdot \exp\left((r_{result-P}+r_{comfort-P})/150,000\right) \end{array}\right.$$

Based on [3], the specific reward functions and penalty functions were designed. Firstly, we designed the corresponding reward function and penalty function based on the results of the driving tasks. The independent variable of the reward function is the number of CAVs and HV2 reaching destinations, which aims to train decisions that can assist HVs in completing driving tasks. The penalty function is designed based on collisions.
(21)$$r_{result-I}=300\left(n_{r1}+n_{r2}\right)$$
(22)$$r_{result-P}=-60,000,if\;collision$$
where $n_{r1}$ is the number of CAVs leaving from “Exit 1”. $n_{r2}$ is the number of CAVs leaving from “Exit 1”.

To train the decision-making model to improve traffic efficiency, this paper divides the speed interval of CAVs into three parts. The corresponding reward and penalty functions were designed to curb speeding, encourage high-speed driving, and punish low-speed blocking for these three-speed ranges. In order to make CAVs faster and more stable to explore the optimal speed, we used the exponential function to design a soft reward function [28].
(23)$$r_{efficiency-I}=\exp\left[\frac{6\times (v_{y}-v_{y\min})}{v_{y\max}}\right],if\;v_{y\min}\leq v_{y}\leq v_{y\max}$$
(24)$$r_{efficiency-P}=\left\{\begin{array}{c} -\exp\left(v_{y}-v_{y\max}\right),if\;\;v_{y}>v_{y\max}\\ -\exp\left(1+\frac{v_{y\min}-v_{y}}{3}\right),if\;v_{y}<v_{y\min} \end{array}\right.$$
where $v_{y}$ is the velocity of the CAV. $v_{y\max}$ represents the maximum velocity allowed by the current lane; its value is 25 $m\cdot s^{-1}$, or 15 $m\cdot s^{-1}$.

In order to improve the ride comfort of all vehicles in this traffic section, the corresponding reward function and penalty function are designed based on the acceleration and lane change times of all vehicles.
(25)$$r_{comfort-I}=\exp\left(3\right)\times n_{c1}$$
(26)$$r_{comfort-P}=-2000\times n_{c2}-\exp\left(\frac{m}{2}\right)$$
where $n_{c1}$ is the number of vehicles with acceleration of $[-2\;m\cdot s^{-2},2\;m\cdot s^{-2}]$. $n_{c2}$ is the number of vehicles with acceleration of $(-\infty ,-4.5\;m\cdot s^{-2}]$. *m* is the number of lane changes in this traffic section within 0.5 s.

Superior security performance is the premise of developing decision technology [29]. In [30], the length of CAVs’ safety time is one of the most important factors affecting road safety. This paper introduces the safety time of the CAVs into the corresponding reward function. The definition of the safety time is as follows: (27)$$t_{1}=t_{safe-follower}=\frac{y_{AV}-y_{follower}}{v_{follower}-v_{AV}}$$
(28)$$t_{2}=t_{safe-leader}=\frac{y_{leader}-y_{AV}}{v_{AV}-v_{leader}}$$
where $y_{AV}$ is the longitudinal position of the CAV. $y_{leader}$ and $y_{follower}$ are the longitudinal positions of the front and rear vehicles of CAV, respectively. $v_{leader}$ and $v_{follower}$ are the longitudinal speeds of the front and rear vehicles of CAV, respectively.

A driving hazard diagram is proposed to represent the degree of danger of the vehicle’s state based on safety time $t_{safe-follower}$ and $t_{safe-leader}$. As shown in Figure 3, three primary colors are used to represent the degrees of danger in this state. The red region represents collision accident danger, and the deeper the color, the greater the likelihood. The yellow area indicates that the vehicle needs to pay attention to the occurrence of a possible emergency. The green area indicates that the vehicle is in a safe state. By dividing Figure 3 into five categories, the sub-reward functions were designed. On the basis of the above principles, a security-based reward function is proposed for training security decisions. The formula is shown in Table 4.

## 5. Experiments

### 5.1. Parameter Setting

In this section, we show the solid interaction scene designed through SUMO. Firstly, based on this scene, the proposed reward function is compared with the traditional reward function. The two algorithms were used to train four decision-making modes, and the differences between different algorithms and modes are compared. Finally, the performances of four decision-making modes based on the two algorithms are evaluated. The main parameters for algorithms are listed in Table 5.

### 5.2. Evaluation Indexes

In order to further evaluate the test performance of each decision-making mode, four kinds of evaluation indexes are proposed, namely, efficiency index, safety index, comfort index, and task index.

1.Efficiency: The average longitudinal velocity of CAVs and all vehicles in this scenario is proposed and used to evaluate the traffic efficiency of CAVs and the comprehensive efficiency of total traffic flow under different decision modes. Their functions are defined as follows:
(29)$${\bar{v}}_{AVs}=\frac{\sum_{i=1}^{n}\sum_{j=1}^{m}v_{AV-ij}}{N}$$
(30)$$\bar{v}s.=\frac{\sum_{i=1}^{n}\sum_{j=1}^{m}v_{ij}}{N_{all}}$$
where $v_{AV-ij}$ represents the longitudinal velocity of the ith CAV detected at the jth time step. $v_{ij}$ is the longitudinal velocity of the ith vehicle detected at the jth time step. *N* indicates the total number of CAVs’ longitudinal velocities detected in all-time steps. $N_{all}$ indicates the total number of all vehicles’ longitudinal velocities detected in all-time steps.2.Safety: Due to the presence of multiple CAVs in the test scenario, we define the collision rate as the probability of each CAV collision accident in each episode.
(31)$$p_{collision}=\frac{N_{collision}}{N_{CAV}}$$
where $N_{collision}$ is the number of collisions in a single episode. $N_{CAV}$ represents the number of CAVs.3.Comfort: For the evaluation of comfort, we mainly studied the deceleration of all vehicles under different decision-making modes. All vehicles’ total emergency braking times $N_{braking}$ in a single episode are defined as comfort evaluation indexes. It should be noted that the deceleration between $(-\infty ,-4.5\;m\cdot s^{-2}]$ is regarded as emergency braking.4.Task: Based on the solid interaction between CAVs and HVs in the test scenario, the driving task completion rates of CAVs and HV2 are used as the task indexes. Their functions are defined as follows.
(32)$$p_{CAV}=\frac{N_{CAV-Exit1}}{N_{CAV}}$$
(33)$$p_{HV2}=\frac{N_{HV2-Exit1}}{N_{HV2}}$$
where $N_{CAV-Exit1}$ is the number of CAVs that left via “Exit 1”. $N_{HV2-Exit1}$ indicates the number of HV2 that left via “Exit 1”. $N_{HV2}$ represents the number of HV2.

### 5.3. Validation of the Reward Function

The proposed reward function was used in GQN and MDGQN to train different decision-making modes while improving training efficiency. In [27], the traditional reward function only considers intention reward, speed reward, lane-changing penalty, and collision penalty. In this article, this traditional reward function is used as the baseline, and the reward of the CONP decision-making mode training process is compared. In order to allow them to make a fair comparison based on the maximum and minimum reward values they can achieve, the reward values in the training process are normalized.

As shown in Figure 4, using the proposed reward function for training can explore the maximum reward rapidly. The traditional reward function’s training process experiences repeated reward reductions, and the maximum reward cannot be explored in the specified number of rounds. In addition, the training stability is greatly improved by comparing the reward fluctuation in the training process with the baseline. After the 200th episode, the average normalized reward values of Baseline-GCQ, CONP-GCQ, and CONP-MDGCQ were 0.7399, 0.9882, and 0.9886, respectively; and their variances were 0.0871, 0.0028, and 0.0049. Under the condition of using the GQN algorithm, the CONP decision-making model was 35.40% better than the average of baseline, and the variance is only 3.27% higher than that of the baseline. In summary, the proposed reward function can promote the fast convergence of the algorithm and greatly enhance the stability of the training process.

### 5.4. Training of Different Decision-Making Modes

As shown in Figure 5, under the four decision-making modes, the MDGQN algorithm converged faster than GQN, and the fluctuation of reward decline decreased significantly. This verified that MDGQN effectively alleviates the overestimation problem after using the multi-step TD target and double Q-learning algorithm to explore the optimal action more quickly. Based on MDGQN algorithm analysis, the AGGI decision-making mode converged in the 80th episode. In contrast, the AGGP decision-making mode did not fully converge until the 160th episode, which further verifies that the reward and punishment ratio of the reward function affects the convergence speed of the algorithm. In summary, by comparing the four decision-making modes, the aggressive decision-making mode uses more punishment than the conservative decision-making mode. The incentive decision-making mode can promote the convergence of the algorithm faster than the punished decision-making mode.

Furthermore, the training decision-making mode was tested, and the reward mean and variance are compared. As shown in Figure 6, MDGQN explored higher reward thresholds and averages than GQN in all four decision modes. Under AGGP, AGGI, CONP, and CONI decision-making modes, the corresponding test reward variances of MDGQN were 93.91, 80.26, 64.71, and 87.10; these were less than 108.51, 108.19, 93.78, and 97.01 for GQN. It can be concluded that the test stability of the CONP decision mode is the strongest, but the ability to distinguish algorithm differences is poor. AGGI is the decision-making mode that can best reflect the differences between algorithms, but the test stability decreases slightly.

### 5.5. Evaluation of Decision-Making Modes

Based on Figure 7, the two algorithms are first compared. In the four modes, the average speed of CAVs was better than that of GQN except for the CONI decision mode. MDGQN performed better than GQN in the speed of total traffic flow under the four modes. However, the proportion of CAVs’ average speed increase in the total traffic flow’s average speed increase is less than 1. In the CONI decision-making mode, the average speed of CAVs trained by MDGQN was lower than that of GQN, but the total traffic flow speed increased. This shows that based on the designed reward function, the MDGQN algorithm can obtain decisions that improve the total traffic efficiency by training CAVs. As shown in Table 6, MDGQN had a lower collision rate, fewer emergency braking times, and a higher arrival rate of CAVs and HV2 than GQN under the same decision-making mode, except for a slight increase in the number of emergency braking incidents under CONP decision-making mode. This can be analyzed from the corresponding data. Under the CONP decision-making mode, the $p_{HV2}$ of MDGQN was 0.717, which was 18.12% higher than that of GQN. When the HV2 increases in “Exit 1”, the uncertainty of the scene will enhance, increasing the number of emergency brake events. From the above results, it can be concluded that the comprehensive performance of the MDGQN algorithm is superior to that of GQN in this specific autonomous driving scene, and the MDGQN is more suitable for solid, interactive, uncertain scenes.

Additionally, the four decision-making modes are compared. Compared with the conservative decision-making mode, the aggressive decision-making mode had a higher average speed, a higher collision rate, more emergency braking incidents, and a higher vehicle arrival rate. Compared with the punished decision-making model, the average speed of the incentive decision-making mode was slightly higher, the collision rate was higher, the number of emergency braking incidents was higher, and the arrival rate of HV2 was further improved. Among the four decision-making modes, AGGI decision-making mode had the highest traffic efficiency; CAVs and HV2 had the highest arrival rate, but their collision rate and number of emergency braking incidents were are also the highest. The collision rate and emergency braking incidents of CONP decision mode were the lowest, but the traffic efficiency was the lowest, and the arrival rates of CAVs and HV2 were the lowest. Those performance differences are consistent with the weight coefficient allocation of the proposed reward function matrix. This fully proves that the proposed reward function matrix can effectively train various decision-making modes to adapt to autonomous driving scenarios.

## 6. Conclusions

We proposed a reward function matrix, including a decision-weighted coefficient matrix, an incentive-punished-weighted coefficient matrix, and a reward–penalty function matrix. By adjusting the weight coefficient of the reward function matrix, various decision-making modes can be trained. We trained four decision-making modes, namely, AGGI, AGGP, CONI, and CONP; and the GQN algorithm and the MDGQN algorithm based on GQN improvement were used for verification by comparison. A large number of simulation results proved the following three conclusions. Firstly, the proposed reward function can promote the fast convergence of the algorithm and greatly improve the stability of the training process. Taking the CONP decision-making mode as an example, the average normalized reward value after the 200th round was 35.40% higher than that of the baseline, and the variance was only 3.27% greater than that of the baseline. Secondly, the comprehensive performance of the MDGQN algorithm is superior to that of GQN. Under the four decision-making modes, the averages and variances of test reward values of MDGQN are better than those of GQN. In terms of driving performance, MDGQN performs better than GQN, except that the number of brakes increases slightly in CONP decision-making mode. Finally, the proposed reward function matrix can effectively train various decision-making modes to adapt to different autonomous driving scenarios. With an increase in incentive weight, the comparison effect of the algorithm is more obvious, but the security will decrease. In our future work, we will further study the interactions of autonomous vehicles and decision mode switching.

## Figures and Tables

**Figure 1 sensors-22-04586-f001:**
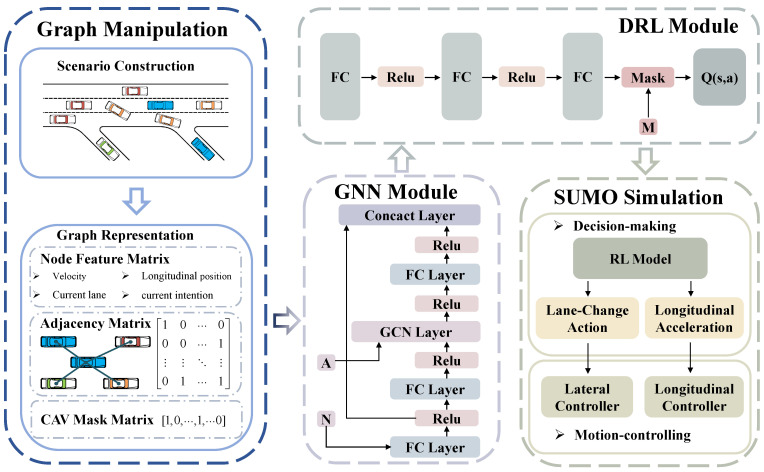
The schematic diagram of the designed framework. Letters N, A, and M represent node feature matrix, adjacency matrix, and CAV mask matrix respectively.

**Figure 2 sensors-22-04586-f002:**
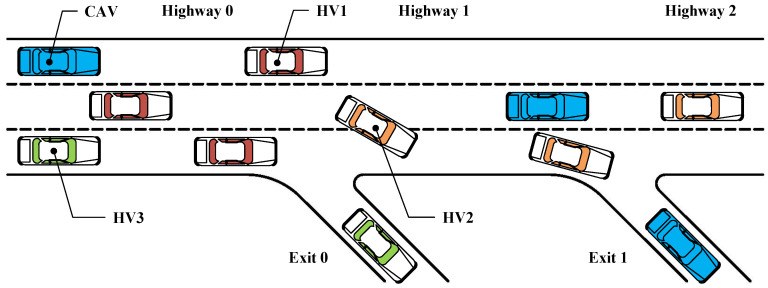
The highway exit scene with solid interactions between CAVs and HVs.

**Figure 3 sensors-22-04586-f003:**
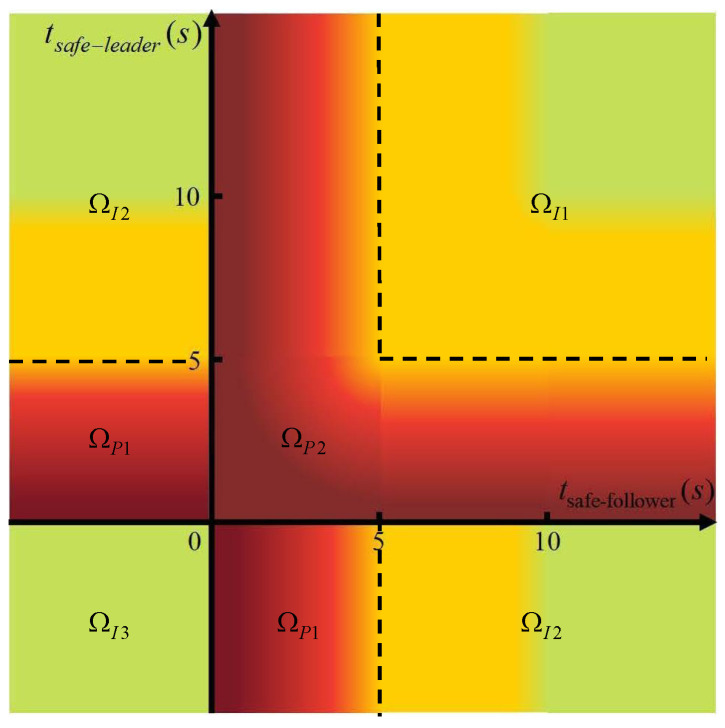
The driving hazard diagram.

**Figure 4 sensors-22-04586-f004:**
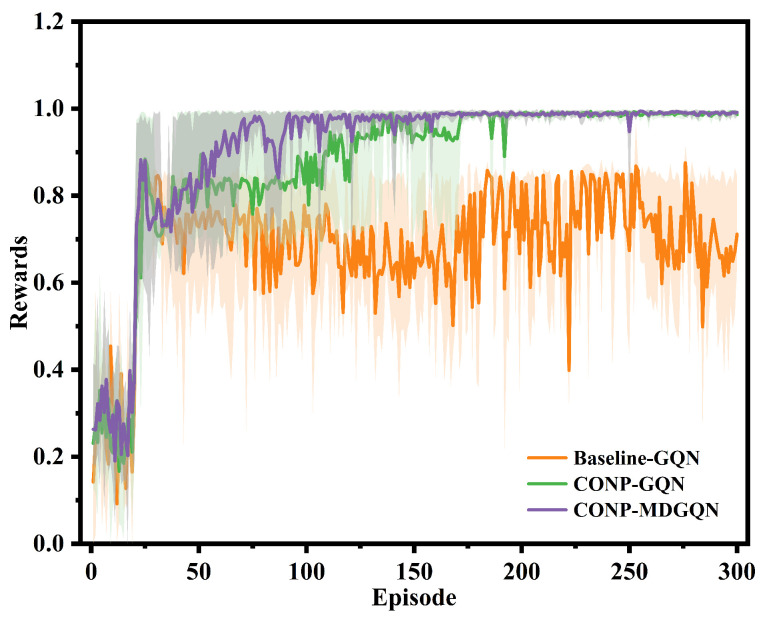
The comparison between the proposed reward function and baseline.

**Figure 5 sensors-22-04586-f005:**
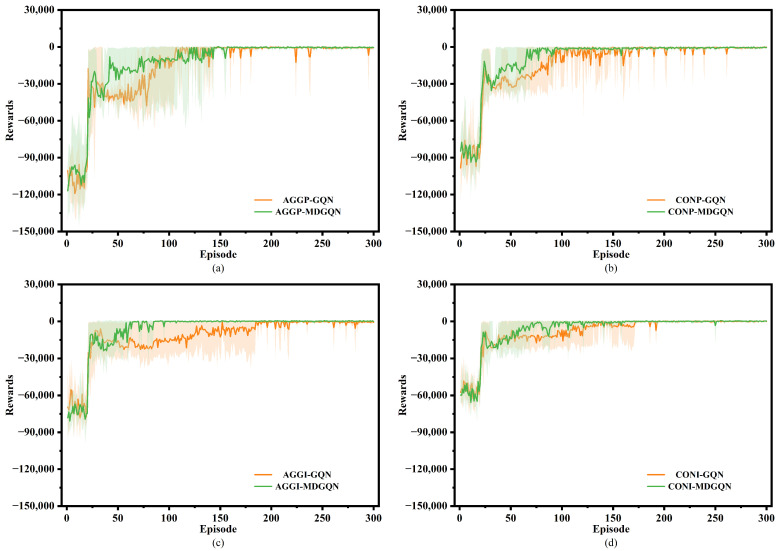
The training reward diagrams of four modes are compared and verified by the GQN and MDGQN algorithms. (**a**) The AGGP decision-making mode, (**b**) the CONP decision-making mode, (**c**) the AGGI decision-making mode, and (**d**) the CONI decision-making mode.

**Figure 6 sensors-22-04586-f006:**
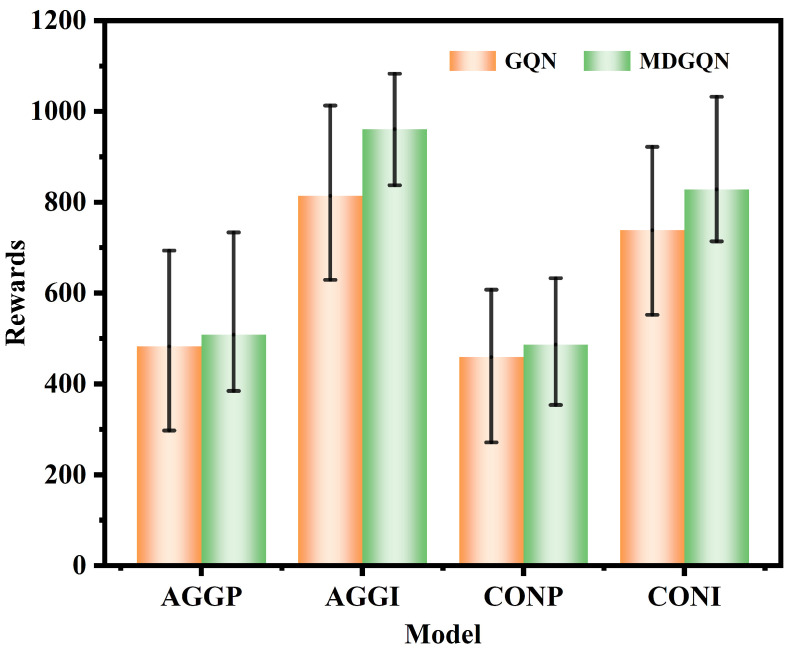
Test reward diagram of four decision-making modes. Orange represents the decision-making mode using the GQN algorithm, and green indicates the decision-making mode using the MDGQN algorithm. In the figure, the two ends of the error rod represent the maximum and minimum values of the test values.

**Figure 7 sensors-22-04586-f007:**
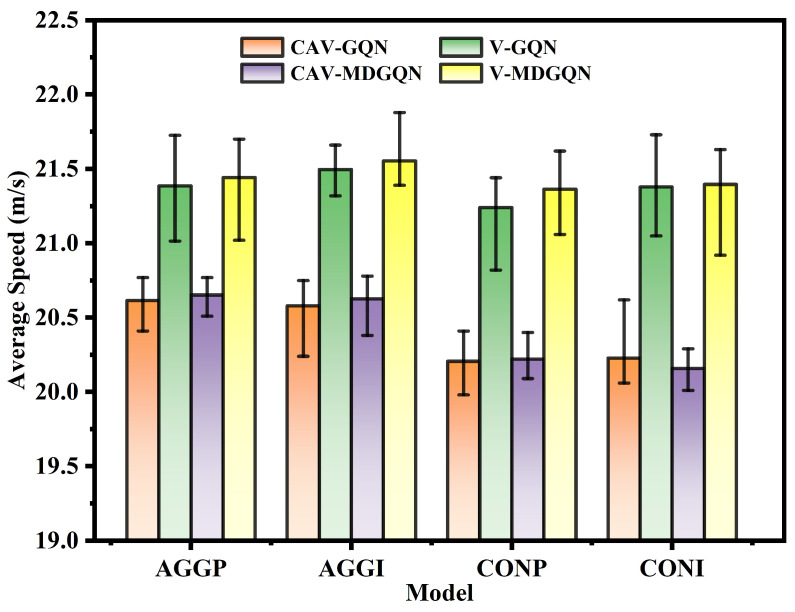
Average longitudinal velocity of CAVs and total traffic flow under four decision-making modes. The CAV-GQN means the average longitudinal speed of CAVs under the training of the GQN algorithm. The CAV-MDGQN represents the average longitudinal velocity of CAVs under the MDGQN algorithm training. The V-GQN indicates the average longitudinal speed of the total traffic flow under the training of the GQN algorithm. The V-MDGQN indicates the average longitudinal velocity of total traffic flow under MDGQN training. The two ends of the error rod indicate the maximum and minimum values of the test values.

**Table 1 sensors-22-04586-t001:** Specific scenario setting parameters.

Parameter	Value	Parameter	Value
Length of “Highway 0”	150 m	Number of HV1	20
Length of “Highway 1”	200 m	Number of HV2	10
Length of “Highway 2”	150 m	Number of HV3	15
Time step	0.1 s	Number of CAVs	15
Initial velocity of HV1	24m/s	Probability of HV1 appearing every 0.1 s	0.18
Initial velocity of HV2	22 m/s	Probability of HV2 appearing every 0.1 s	0.08
Initial velocity of HV3	20 m/s	Probability of HV3 appearing every 0.1 s	0.12
nitial velocity of CAVs	20 m/s	Probability of CAVs appearing every 0.1 s	0.12

**Table 2 sensors-22-04586-t002:** Weight coefficient of the decision-weighted coefficient matrix.

Parameters	Conservative	Aggressive
The weight of task $k_{r}$	0.2	0.25
The weight of efficiency $k_{e}$	0.2	0.35
The weight of comfort $k_{c}$	0.25	0.15
The weight of safety $k_{s}$	0.35	0.25

**Table 3 sensors-22-04586-t003:** Weight coefficient of the incentive-punished-weighted coefficient matrix.

Parameters	Incentive	Punished
The weight of incentive sub-reward function $k_{rI0}$, $k_{eI0}$, $k_{cI0}$, $k_{sI0}$	0.6	0.4
The weight of punished sub-reward function.$k_{rP0}$, $k_{eP0}$, $k_{cP0}$, $k_{sP0}$	0.4	0.6

**Table 4 sensors-22-04586-t004:** The security-based reward function.

Subfunction	Category	Calculation
$r_{safe-I}$	$\Omega_{I1}$	$\max\left(\exp\left(\frac{\min(t_{1},t_{2})-5}{2.5}\right),\exp\left(2\right)\right)$
	$\Omega_{I2}$	$\max\left(\exp\left(\frac{\max(t_{1},t_{2})-5}{2.5}\right),\exp\left(2\right)\right)$
	$\Omega_{I3}$	exp(2)
$r_{safe-P}$	$\Omega_{P1}$	$-\exp(4-\min\left(t_{1},t_{2}\right))$
	$\Omega_{P2}$	$-\exp(4-\max\left(t_{1},t_{2}\right))$

**Table 5 sensors-22-04586-t005:** Parameters of the graph reinforcement learning.

Parameters	Value
Optimizer	Adam
Nonlinearity	Relu
Learning rate	0.005
Discount factor	0.99
Updating rate	0.05
Minibatch size	32
Multi-step	3

**Table 6 sensors-22-04586-t006:** Test evaluation results.

Modes	Algorithm	$p_{\mathrm{collision}}$	$N_{\mathrm{braking}}$	$p_{\mathrm{CAV}}$	$p_{\mathrm{HV}2}$
**AGGP**	GQN	0.0111	3.1	0.991	0.763
	MDGQN	0.0067	1.867	1	0.813
**AGGI**	GQN	0.0133	3.567	1	0.93
	MDGQN	0.0022	2.533	1	0.937
**CONP**	GQN	0.0045	1.667	0.987	0.607
	MDGQN	0.0	1.9	0.996	0.717
**CONI**	GQN	0.0111	2.5	1	0.857
	MDGQN	0.0045	2.3	1	0.83

## Data Availability

Not applicable.

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
