# Peer review of "Multi-Agent Decision-Making Modes in Uncertain Interactive Traffic Scenarios via Graph Convolution-Based Deep Reinforcement Learning"

_sensors, 2022, doi:10.3390/s22124586_

Round 1

Reviewer 1 Report

Dear Authors,

The work concerns the application of deep learning in the flow control of autonomous and traditionally human-controlled vehicles, which is quite a challenge for road safety. A reward function matrix has been proposed to train different decision-making modes with a different emphasis on decision-making styles and a further emphasis on incentives and punishments. Traffic modeling using a graph model was also presented to better map the interactions between vehicles. Overall, the work is very interesting and brings a new contribution to the presented issues. Unfortunately, there are some editing errors in the work that should be corrected, moreover, the following issues require clarification:

1. In my opinion, the introduction and references (only 18) are unfortunately too limited. This section should be expanded with information related to autonomous vehicles and transport safety, which is directly related to making the right decisions in road traffic.

2. Table 1 should rather be placed on page 4 in chapter 2.1. (line 143).

3. The work methodology is sufficiently described.

4. Unfortunately, there are no comments - the experimental part of the discussion on the results obtained by other researchers should be supplemented. It is not best to end the chapter with a drawing or table without comment.

I think the following references should be taken into account by authors in the manuscript:

https://doi.org/10.3390/app12062993

https://doi.org/10.3390/en14164777

https://doi.org/10.1515/eng-2020-0006

https://doi.org/10.1007/978‐3‐319‐44427‐7_24

https://doi.org/10.14669/AM.VOL86.ART2

https://doi.org/10.1109/TRO.2016.2624754

https://doi.org/10.1007/978-3-030-18963-1_2

https://doi.org/10.1016/j.trpro.2020.02.053

https://doi.org/10.1016/j.ifacol.2018.08.315

https://doi.org/10.20858/sjsutst.2018.100.2

https://doi.org/10.3390/su13126654.

https://doi.org/10.26552/com.C.2021.2.F33-F42

https://doi.org/10.3390/su13084443

10.1016 / j.compenvurbsys. 2018.11.007.

Thank you.

Reviewer 2 Report

The manuscript introduces a reward function matrix, including a decision-weighted coefficient matrix, an incentive-punished-weighted coefficient matrix, and a reward-penalty function matrix for the fast convergence of the algorithm, and greatly enhance the stability of the training process. Simulated results were given and analysed in light of the MDGQN algorithm's convergence and performance in autonomous driving scenarios when compared to the GQN. Furthermore, different decision-making modes can be tailored to meet the needs of various driving tasks.

I think the study is interesting, and the results and findings offered will be valuable to the audience.

I've included my suggestions and comments below for the authors to consider as they work to improve the manuscript's quality. It is noted that the multi-agent decision-making modes in uncertain interactive traffic scenarios have been established earlier, as the authors acknowledged this through well-known references. I have looked for the originality and Innovation in this study, and I suggest the authors emphasise this matter. I noted on page 3 (lines: 89-111) presents the main contributions; however, It is not clear to me about the Innovation of this study.

Introduction: There is a lack of in-depth review in the area of research where there have only 18 papers been reviewed. I suggest the authors carry out a comprehensive and critical review on the research area and clearly determine a Research Gap.

Tables 2, 3 & 4. I am not convinced how were these parameters selected? I suggest the authors to elaborate more on the derivation of these parameters. 

I suggest proper references/original sources for Equations 20-26.

Figure 4: there is a lack of in-depth data interpretation on the data presented in this Figure; and thereby, I suggest the authors attend on this matter. 

It is not clear to me how were Equations 29 &30 derived and applied for training security decisions.

Conclusions: I suggest revising the Conclusions, covering maim findings/outcomes of the current study.

Round 2

Reviewer 1 Report

Dear Authors,

thanks for major revisions to the manuscript. The introductory part has been supplemented and the conclusions have been corrected, the work is much clearer. I would also like to thank the authors for their answers, I accept the changes made.

In view of the above, I recommend the work to be printed in its current form.

Thank you.

Reviewer 2 Report

The authors have attended all comment and suggestion provided by the reviewer. I have now recommended for the acceptance of the revised manuscript.